# LLM-CoT Enhanced Graph Neural Recommendation with Harmonized Group Policy Optimization

## Abstract

Graph neural networks (GNNs) have advanced recommender systems by modeling interaction relationships. However, existing graph-based recommenders rely on sparse ID features and do not fully exploit textual information, resulting in low information density within representations. Furthermore, graph contrastive learning faces challenges. Random negative sampling can introduce false negative samples, while fixed temperature coefficients cannot adapt to the heterogeneity of different nodes. In addition, current efforts to enhance recommendations with large language models (LLMs) have not fully utilized their Chain-of-Thought (CoT) reasoning capabilities to guide representation learning. To address these limitations, we introduces LGHRec (LLM-CoT Enhanced Graph Neural Recommendation with Harmonized Group Policy Optimization). This framework leverages the CoT reasoning ability of LLMs to generate semantic IDs, enriching reasoning processes and improving information density and semantic quality of representations. Moreover, we design a reinforcement learning algorithm, Harmonized Group Policy Optimization (HGPO), to optimize negative sampling strategies and temperature coefficients in contrastive learning. This approach enhances long-tail recommendation performance and ensures optimization consistency across different groups. Experimental results on three datasets demonstrate that LGHRec improves representation quality through semantic IDs generated by LLM's CoT reasoning and effectively boosts contrastive learning with HGPO. Our method outperforms several baseline models. The code is available at: `https://anonymous.4open.science/r/LLM-Rec`.

## 1 Introduction

Recently, LLMs Guo et al. (2025); Wei et al. (2022) have advanced the recommendation community Lin et al. (2024); Kaur et al. (2025); Zhang et al. (2025a). Their generative capabilities enable the provision of rich semantic information, forming a one-stage recommendation paradigm. This paradigm shows promise in addressing the information loss issue that arises in traditional multi stage recommender systems.Current research on LLMs for recommendation can be divided into two categories. The first approach treats LLMs as recommender systems (LLMs as RSs) Chen et al. (2024); Yin et al. (2023); Zheng et al. (2024); Wang et al. (2019). However, it faces challenges such as low online inference efficiency, insufficient use of collaborative filtering signals, and hallucination issues Ji et al. (2023); Yao et al. (2023). The second approach utilizes knowledge generated by LLMs to enhance existing models (LLM-enhanced RSs) Hu et al. (2025); Wang et al. (2024); Yang et al. (2024); Liu et al. (2024a); Ren et al. (2024). This approach is more flexible but has not fully explored the deep potential of LLMs, particularly their CoT reasoning abilities.Most methods mainly rely on information extracted from general domain knowledge, neglecting the possibility of guiding LLMs to perform deeper semantic reasoning for recommendation tasks. How to guide LLMs to leverage CoT reasoning capabilities and enhance collaborative filtering signals remains an unresolved challenge.

GNNs Wang et al. (2019); He et al. (2020); Wu et al. (2022; 2024a) can capture higher-order collaborative signals but have drawbacks. They rely on ID features and struggle to leverage rich textual information for semantic modeling, resulting in insufficient information density in representations, particularly for long-tail items, where the representation quality is poor. GNNs are also sensitive to

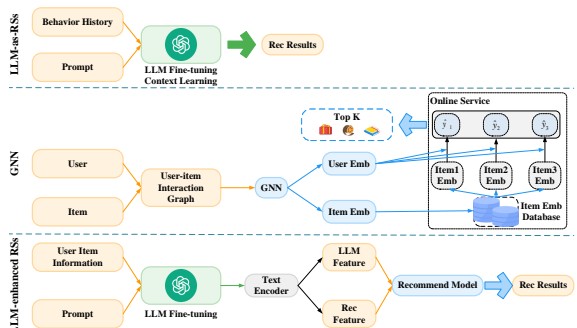

Figure 1: The differences between the three recommendation paradigms.

data sparsity Cao et al. (2023) and introduce noise when aggregating high-order neighbor information Jiang et al. (2023). Therefore, researchers introduce graph contrastive learning Lin et al. (2022), which enhance representation learning through structural and semantic contrastive losses. However, existing graph contrastive learning have limitations as well. In large scale scenarios, their random negative sampling introduces false negatives, which can mislead model's optimization. Additionally, the fixed temperature coefficient in the contrastive loss is not adaptable to the varying embedding characteristics of groups with different degrees. This leads to poor contrastive learning results, particularly for long-tail items. The differences of three paradigms are illustrated in Figure 1.

To address these issues, we propose LGHRec, it integrates CoT reasoning ability of LLM with reinforcement learning for collaborative optimization. The goal is to combine semantic reasoning capabilities of LLMs with the collaborative filter strengths of GNNs. Reinforcement learning is used to optimize graph contrastive learning and enhance representation quality. LGHRec is LLM-enhanced RSs paradigm. Offline, CoT reasoning of LLM generates item descriptions and extracts embeddings as semantic IDs with higher information density. These semantic IDs are fused with IDs during training to serve as initial item representations for the GNN. This allows the GNN to learn higher quality representations by utilizing deep semantic information. Since semantic IDs are stored offline, the delay from online LLM calls is avoided, making LGHRec suitable for industrial applications. To address challenges in contrastive learning, including optimal negative sampling and temperature coefficient selection across groups with varying degrees, as well as performance imbalance in Group Relative Policy Optimization (GRPO) Shao et al. (2024), we introduce HGPO algorithm. HGPO incorporates cross-group coordination mechanism that constrains strategy differences between groups, ensure global strategy consistency while adapt to the characteristics of each group. It improves long-tail item recommendation performance.The contributions are as follows:

- We propose LGHRec, which leverages the CoT reasoning capabilities of LLMs to generate high quality semantic IDs for GNNs. This approach enhances the information density of ID features in GNNs while avoiding the high computational cost of online LLM inference.

- We introduce the HGPO algorithm to optimize graph contrastive learning. By employing adaptive negative sampling, temperature coefficient adjustments, and a cross-group coordination mechanism, HGPO improves contrastive learning performance, enhances the model's adaptability to heterogeneous data, and boosts long-tail recommendation performance.

- We conduct extensive experiments on three datasets, demonstrating that LGHRec outperforms several baseline models and validating the effectiveness of the proposed method.

## 2 METHODS

We introduce the Deep Semantic Embedding Generator (DSEG) and HGPO. The implementation details of the GNN are provided in the Appendix. The architecture of LGHRec is shown in Figure 2.

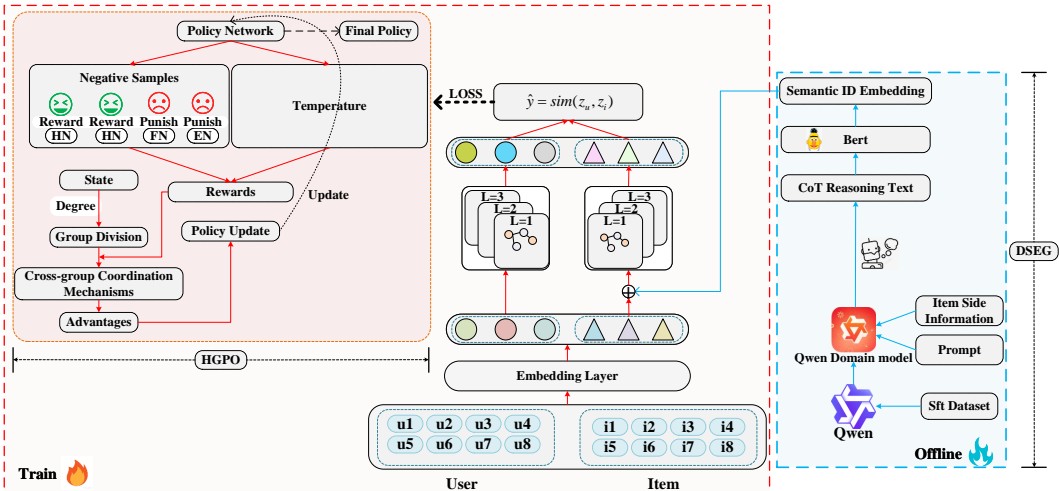

Figure 2: The architecture diagram of the proposed LGHRec.

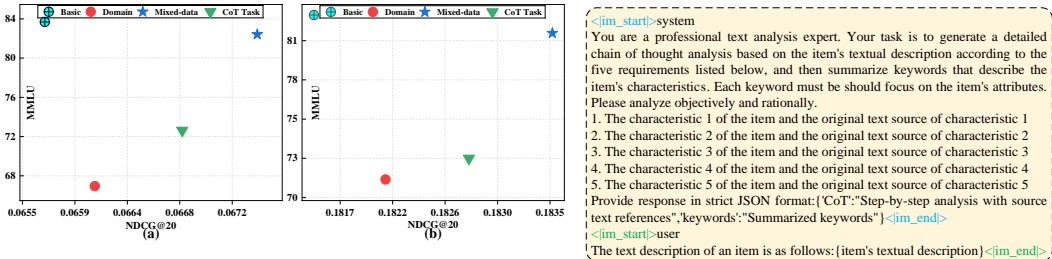

Figure 3: After using various fine-tuning methods, the general capabilities and recommendation performance of LGHRec: (a)Yelp dataset, (b)MIND dataset.

Figure 4: The prompt template for guiding LLM to perform CoT reasoning.

## 2.1 DEEP SEMANTIC EMBEDDING GENERATOR

Fine-tuning LLM can improve performance Bao et al. (2023). We explore some fine-tuning methods to generate item CoT reasoning text. We evaluate the NDCG@20 and MMLU, using Qwen2.5-32B-Instruct model under various fine-tuning methods, including base, domain-adaptive, CoT task and mixed fine-tuning. Mixed fine-tuning, which combines recommendation CoT dataset with general dataset, helps retain the model's foundational knowledge and prevents catastrophic forgetting. As shown in Figure 3, this methods achieves the best balance between recommendation performance and general capabilities, making it the preferred method for LGHRec. We designed prompts, as shown in Figure 4, to guide LLM CoT reasoning, denoted as $T_{CoT}^{(i)}$, which is then encoded into semantic IDs $\mathbf{e}_{CoT}^{(i)} \in \mathbb{R}^{d_c}$ using BERT model. This offline process ensures that item is processed once and periodically updated, storing semantic IDs for direct use during GNN training. It leverages the LLM's reasoning capabilities while avoiding the latency of online services. To fuse semantic IDs with collaborative filtering signals, we concatenate them with the ID embeddings $\mathbf{e}_{ID}^{(i)} \in \mathbb{R}^{d_{id}}$ and apply linear layer, as initial item representations $\mathbf{e}_i^{(0)}$. The initial user representation $\mathbf{e}_u^{(0)}$ is based on ID embeddings, because user behavior changes more frequently, require real-time updates.

## 2.2 DEFINITION OF REINFORCEMENT LEARNING

In large scale recommender systems, contrastive learning face two challenges. First, calculate similarity of all samples is expensive, and random sampling may introduce false negatives. So, selecting high quality negative samples is essential. Second, the fixed temperature coefficient $\tau$ cannot adapt to the heterogeneity of users and items. Active users or popular items with higher node degrees require smaller $\tau$ to enhance the distinction of hard negative samples, while low activity users

or long-tail items with sparse information need larger $\tau$ to stabilize learn. To address these problems, we model the optimization of contrastive learning as reinforcement learning problem. The state for user $u$ includes the user's embedding $z_u$, the positive sample embedding $z_k$, the candidate negative sample pool $N_u$, and the user's degree $d_u$. The action consists of selecting $M$ negative samples from $N_u$ and choosing $\tau$ for the anchor. The policy network $\pi_\theta\left(a_t|s_t\right)$ outputs probability of selecting negative samples and $\tau$. The reward reflects the quality of the selected negative samples and $\tau$.

### 2.3 RULE-BASED REWARDS

**Reward Hard Negatives.** Hard negatives are similar to the anchor but should not be mistaken for positive samples. They provide gradient signals, helping the model learn finer features. Therefore, we assign a reward to negative samples that are similar to the anchor user embedding $z_u^{(0)}$ but differ in similarity from the positive sample embedding $z_u^{(k)}$. The expression is as follows:

$$R_{\text{hard}}(z_n^*) = \begin{cases} +w_1 & \text{if } \theta_{\text{easy}} < \text{sim}(z_u^{(0)}, z_n^*) < \theta_{\text{FN}} \text{ and } \text{sim}(z_u^{(k)}, z_n^*) < \theta_{\text{FP}} \\ 0 & \text{otherwise} \end{cases} \tag{1}$$

where $\theta_{\text{FN}}, \theta_{\text{easy}}, \theta_{\text{FP}}$ are similarity thresholds, and $w_1 > 0$.

**Punish False Negatives.** False negatives are actually similar to the anchor but are incorrectly selected as negative samples. If the model treats them as negatives, it will mislead the learning process. Therefore, we assign a negative reward to samples that are highly similar to either the anchor user embedding $z_u^{(0)}$ or the positive sample embedding $z_u^{(k)}$. The reward is as follows:

$$R_{\text{false}}(z_n^*) = \begin{cases} -w_2 & \text{if } \text{sim}(z_u^{(0)}, z_n^*) \geq \theta_{\text{FN}} \\ -w_3 & \text{if } \text{sim}(z_u^{(k)}, z_n^*) \geq \theta_{\text{FP}} \\ 0 & \text{otherwise} \end{cases} \tag{2}$$

**Punish Easy Negatives.** Easy negatives are very dissimilar to the anchor, making them easy for the model to distinguish. The gradient information they provide is limited, and their contribution to the model's learning is minimal. If the model frequently selects easy negatives, it may become less effective during training, failing to fully utilize hard negatives that enhance its discriminative ability. Therefore, we assign a negative reward to samples that are very dissimilar to the anchor user embedding $z_u^{(0)}$. The reward is defined as follows:

$$R_{\text{easy}}(z_n^*) = \begin{cases} -w_4 & \text{if } \text{sim}\left(z_u^{(0)}, z_n^*\right) \leq \theta_{\text{easy\_low}} \\ 0 & \text{otherwise} \end{cases} \tag{3}$$

Where, $\theta_{\text{easy\_low}}$ is a similarity threshold. The total reward for negative samples is as follows:

$$R_t = R_{\text{hard}} + R_{\text{false}} + R_{\text{easy}} \tag{4}$$

where $w_1 = w_2 = w_3 = 1, w_4 = 0.5$.

**Self-Adaptive Temperature Reward** $R_\tau$**.** In GNNs, nodes with high degrees have rich neighborhood information, making it easier to encounter hard negative samples during contrastive learning. For these nodes, smaller temperature coefficient $\tau$ can amplify similarity differences and help model learn more refined features. Conversely, nodes with low degrees have sparse interactions. The positive samples generated through data augmentation contain noise, result in low similarity with anchor. In this case, smaller $\tau$ would excessively penalize these nodes, hinder the model's ability to learn from sparse positive signals. A larger $\tau$ helps tolerate noise and stabilizes learn for such nodes. So, fixed $\tau$ is insufficient for optimal learning across different node types, and an adaptive mechanism is needed to adjust the strength of contrastive learning based on node degree. We design reward to guide policy network adjust $\tau$ to match the target temperature $T_{\text{ideal}}(d_u)$ according to the degree of the node:

$$R_\tau\left(u, \tau_u^{(t)}\right) = -w_5 \left| \tau_u^{(t)} - T_{\text{ideal}}\left(d_u\right) \right| \tag{5}$$

where $w_5$ is the hyperparameter and $T_{\text{ideal}}(d_u)$ is as follows:

$$T_{\text{ideal}}\left(d_u\right) = \frac{1}{1 + \log\left(1 + d_u\right)} \tag{6}$$

## 2.4 HGPO Mechanism

Existing contrastive learning methods struggle to adapt to all user and item groups, result in insufficient representation learning for low activity users and long-tail items. The GRPO only improves relative performance within a group, which can cause conflicts between strategies across different groups and negative affect long-tail items. So, we propose HGPO, a reinforcement learning algorithm that uses group average rewards to guide policy learning. HGPO introduces a cross-group coordination mechanism to optimize contrastive learning, ensuring global policy consistency while adapting to the unique characteristics of different groups. The mechanism of HGPO is as follows:

**Group Division** $\mathcal{G}$. Nodes are divided into $K$ groups $\mathcal{G} = \{g_1, \ldots, g_K\}$ based degree of the nodes.

**Group Average Reward** $\bar{R}_{g(s_t)}$. For state $s_t$ belonging to group $g$, its group average reward is the expected reward of all possible actions in that state. We estimate it in the train batch $B$ as follows:

$$\bar{R}_g \approx \frac{1}{|B_g|} \sum_{(s'_t, a'_t, r'_t) \in B_g} r'_t \tag{7}$$

where $B_g$ is the set of all samples $(s'_t, a'_t, r'_t)$ in batch $B$ that belong to group $g$. $\bar{R}_g$ represents the average reward level of group $g$ under the current policy.

**Relative Advantage** $A_t^{rel}$. For a sample $(s_t, a_t, r_t)$, its relative advantage is defined as the difference between the actual reward $r_t$ of the action and the group's average reward $\bar{R}_g$: $A_t^{rel} = r_t - \bar{R}_g$. If $A_t^{rel} > 0$, the action outperforms the group average and should be encouraged.

## 2.5 HGPO Objective Function

To maximize relative advantage, add entropy regularization to encourage exploration and introduce a coordination loss to ensure cross-group consistency. The objective function of HGPO is as follows:

$$L_{HGPO}(\theta) = -L^{POLICY}(\theta) + c_1 S[\pi_\theta] + L^{HARM}(\theta) \tag{8}$$

**Policy Loss.** This is a policy gradient term based on the relative advantage $A_t^{\text{rel}}$, and stability is maintained through clipping. The expression is as follows:

$$L^{POLICY}(\theta) = \widehat{\mathbb{E}}_t \left[ \min \left( r_t(\theta) A_t^{\text{rel}}, \text{clip}(r_t(\theta), 1 - \epsilon, 1 + \epsilon) A_t^{\text{rel}} \right) \right] \tag{9}$$

Where, $r_t(\theta) = \frac{\pi_\theta(a_t|s_t)}{\pi_{\theta\text{old}}(a_t|s_t)}$ is the probability ratio between the new and old policies. Maximizing $L^{POLICY}(\theta)$ increases the probability of selecting positive relative advantage actions.

**Entropy Regularization** $S[\pi_\theta]$. Without sufficient exploration, the algorithm tends to converge quickly on groups with more samples or stronger reward signals, such as high activity users and popular items. As a result, long-tail items and low-activity users receive insufficient exploration, leading to poor performance on these groups. Entropy regularization encourages the policy network to maintain randomness, discover customized strategies for different groups, and improve performance on long-tail recommendations. The expression is as follows:

$$S[\pi_\theta] = \widehat{\mathbb{E}}_t[H(\pi_\theta(\cdot \mid s_t))] \tag{10}$$

Since HGPO involves two types of action spaces—negative sample selection and temperature coefficient selection. Therefore, the overall expression is the sum of both:

$$H(\pi_\theta(\cdot \mid s_t)) = H_{neg}(\pi_\theta(a_{neg} \mid s_t)) + H_{temp}(\pi_\theta(a_{temp} \mid s_t)) \tag{11}$$

Where, $H_{neg}$ is the entropy of the negative sample selection action. The policy $\pi_\theta(a_{neg}|s_t)$ outputs a discrete probability distribution $P = \{p_1, p_2, \ldots, p_M\}$ over the $M$ possible negative samples, where $p_j$ is the probability of selecting the $j$-th action, and $\sum_{j=1}^{M} p_j = 1$. The expression is as follows:

$$H_{neg} = -\sum_{j=1}^{M} p_j \log p_j \tag{12}$$

---

**Algorithm 1** HGPO Optimization Process

---

1: **Initialize** policy network parameters $\theta$
2: **for** each training iteration **do**
3:     Collect data $(s_t, a_t, r_t)$ using policy $\pi_\theta$
4:     Divide nodes into groups $\mathcal{G} = \{g_1, \ldots, g_K\}$ based on degrees
5:     **for** each group $g \in \mathcal{G}$ **do**
6:         Calculate group average reward $\bar{R}_g$ from current batch
7:     **end for**
8:     **for** each sample $(s_t, a_t, r_t)$ **do**
9:         Determine group $g$ of state $s_t$
10:        Calculate relative advantage $A_t^{rel} = r_t - \bar{R}_g$
11:     **end for**
12:     Compute policy loss $L^{POLICY}(\theta)$ using relative advantages with clipping
13:     Compute entropy regularization $S[\pi_\theta]$ for negative sampling and temperature selection
14:     Compute harmonizing loss $L^{HARM}(\theta)$ by minimizing variance of group rewards
15:     Update policy network parameters $\theta$ by minimizing total loss:
16:     $L_{HGPO}(\theta) = -L^{POLICY}(\theta) + c_1 S[\pi_\theta] + L^{HARM}(\theta)$
17: **end for**

---

$H_{temp}$ is the entropy of the temperature selection action. The policy $\pi_\theta(a_{temp}|s_t)$ outputs the parameters of a Gaussian distribution for the temperature coefficient in the current state $s_t$, specifically the mean $\mu$ and variance $\sigma^2$, from which the temperature $\mathcal{T}$ is sampled. The expression is as follows:

$$H_{temp} = \frac{1}{2}\log(2\pi e\sigma^2) = \frac{1}{2}\left(1 + \log(2\pi\sigma^2)\right) \tag{13}$$

Therefore, a larger variance results in greater entropy and stronger exploration.

**Coordination Loss $L^{\mathbf{HARM}}(\theta)$.** We minimize the variance of the group average rewards $\bar{R}_g$ using the objective function $L^{HARM}(\theta) = \lambda_{\text{harm}} \cdot \text{Var}_{g \in \mathcal{G}}[\bar{R}_g]$, where $\lambda_{\text{harm}}$ controls the strength of coordination. Minimizing $L^{HARM}$ encourages the policy network to optimize globally while adapting to the characteristics of each group through the relative advantage $A_t^{\text{rel}}$, ensuring similar average reward levels across groups. This approach prevents the algorithm from over optimizing one group at the expense of others. The optimization process of the HGPO algorithm is shown in Algorithm 1.

## 3 EXPERIMENT

### 3.1 OVERALL PERFORMANCE

We applied LGHRec to some baselines across three datasets, with the results shown in Table 1. LGHRec improved the performance of all models. On the sparse Yelp2018 and Amazon-Book datasets, LGHRec mitigated data sparsity challenges through deep semantic augmentation and optimization for long-tail items. This resulted in significant performance improvements of 3% to 7%. Notably, on the denser MIND dataset, where baselines already captured strong collaborative signals, LGHRec still achieved a performance gain of up to 7.49% through its advanced optimization strategies. It demonstrate the robustness of the LGHRec across diverse data environments.

### 3.2 HGPO IN-DEPTH ANALYSIS

**Performance Comparison of Interactive Sparsity Levels.** We divided users and items into five levels based on interaction frequency across the three datasets. We then compared the NDCG@20 performance of LGHRec and the baseline across different activity groups. As shown in Figure 5, LGHRec improved performance for low activity users and reduced the performance gap between groups with varying activity levels. On the Yelp and Book datasets, where long-tail items are more prevalent, LGHRec achieved more substantial improvements, demonstrating its effectiveness in enhancing long-tail recommendations. This improvement is attributed to the HGPO mechanism, which stabilizes the learning of low degree nodes through adaptive temperature adjustment and ensures that long-tail groups are not overshadowed by high activity groups via coordination loss.

**Embedding Distribution Analysis.** We used kernel density estimation to visualize the learned item embeddings on the Yelp dataset, as shown in Figure 6. The results show that embeddings learned

Table 1: Overall performance comparisons.

| Model | | Yelp2018 | | | | Amazon-Book | | | | MIND | | | |
|---|---|---|---|---|---|---|---|---|---|---|---|---|---|
| Baseline | Variants | Recall@10 | Recall@20 | NDCG@10 | NDCG@20 | Recall@10 | Recall@20 | NDCG@10 | NDCG@20 | Recall@10 | Recall@20 | NDCG@10 | NDCG@20 |
| SGL | Base | 0.0363 | 0.0675 | 0.0412 | 0.0555 | 0.0232 | 0.0478 | 0.0306 | 0.0379 | 0.0794 | 0.1366 | 0.0946 | 0.1252 |
| | + LGHRec | 0.0387 | 0.0710 | 0.0428 | 0.0582 | 0.0241 | 0.0508 | 0.0326 | 0.0392 | 0.0826 | 0.1435 | 0.0996 | 0.1301 |
| | RelaImpr ↑ | 6.70% | 5.13% | 3.95% | 4.78% | 3.75% | 6.26% | 6.41% | 3.32% | 3.98% | 5.07% | 5.27% | 3.92% |
| SimGCL | Base | 0.0412 | 0.0721 | 0.0467 | 0.0601 | 0.0248 | 0.0515 | 0.0324 | 0.0410 | 0.0957 | 0.1642 | 0.1012 | 0.1279 |
| | + LGHRec | 0.0430 | 0.0764 | 0.0493 | 0.0620 | 0.0262 | 0.0551 | 0.0345 | 0.0434 | 0.1002 | 0.1680 | 0.1086 | 0.1308 |
| | RelaImpr ↑ | 4.39% | 6.01% | 5.59% | 3.13% | 5.67% | 6.96% | 6.61% | 5.93% | 4.70% | 2.34% | **7.27%** | 2.27% |
| LightGCL | Base | 0.0464 | 0.0793 | 0.0521 | 0.0668 | 0.0312 | 0.0585 | 0.0329 | 0.0436 | 0.1069 | 0.1757 | 0.1134 | 0.1384 |
| | + LGHRec | 0.0480 | 0.0852 | 0.0558 | 0.0692 | 0.0322 | 0.0626 | 0.0346 | 0.0456 | 0.1110 | 0.1829 | 0.1188 | 0.1427 |
| | RelaImpr ↑ | 3.48% | **7.43%** | **7.06%** | 3.53% | 3.28% | 6.94% | 5.19% | 4.65% | 3.80% | 4.10% | 4.78% | 3.14% |
| VGCL | Base | 0.0431 | 0.0757 | 0.0482 | 0.0642 | 0.0278 | 0.0611 | 0.0374 | 0.0476 | 0.1125 | 0.1813 | 0.1187 | 0.1439 |
| | + LGHRec | 0.0453 | 0.0796 | 0.0513 | 0.0689 | 0.0296 | 0.0637 | 0.0392 | 0.0509 | 0.1205 | 0.1851 | 0.1222 | 0.1518 |
| | RelaImpr ↑ | 5.01% | 5.10% | 6.47% | **7.37%** | 6.63% | 4.21% | 4.91% | **6.99%** | 7.14% | 2.08% | 2.95% | 5.47% |
| NESCL | Base | 0.0375 | 0.0743 | 0.0475 | 0.0611 | 0.0298 | 0.0624 | 0.0428 | 0.0513 | 0.1248 | 0.1975 | 0.1366 | 0.1615 |
| | + LGHRec | 0.0390 | 0.0781 | 0.0508 | 0.0633 | 0.0309 | 0.0669 | 0.0456 | 0.0543 | 0.1341 | 0.2101 | 0.1463 | 0.1665 |
| | RelaImpr ↑ | 3.95% | 5.07% | 6.90% | 3.53% | 3.71% | **7.14%** | 6.44% | 5.91% | **7.49%** | 6.35% | 7.12% | 3.12% |
| SCCF | Base | 0.0481 | 0.0799 | 0.0474 | 0.0638 | 0.0337 | 0.0639 | 0.0438 | 0.0522 | 0.1203 | 0.1879 | 0.1295 | 0.1742 |
| | + LGHRec | 0.0498 | 0.0850 | 0.0500 | 0.0671 | 0.0350 | 0.0680 | 0.0461 | 0.0556 | 0.1240 | 0.2004 | 0.1324 | 0.1813 |
| | RelaImpr ↑ | 3.51% | 6.39% | 5.38% | 5.24% | 3.92% | 6.44% | 5.24% | 6.51% | 3.09% | **6.64%** | 2.23% | 4.05% |
| CIKG | Base | 0.0469 | 0.0761 | 0.0465 | 0.0613 | 0.0659 | 0.1077 | 0.0747 | 0.0921 | 0.1256 | 0.1889 | 0.1312 | 0.1631 |
| | + LGHRec | 0.0495 | 0.0808 | 0.0487 | 0.0650 | 0.0675 | 0.1152 | 0.0794 | 0.0972 | 0.1314 | 0.1958 | 0.1349 | 0.1725 |
| | RelaImpr ↑ | 5.53% | 6.13% | 4.83% | 6.01% | 2.47% | 6.93% | 6.24% | 5.53% | 4.60% | 3.66% | 2.81% | **5.75%** |
| AutoGraph | Base | 0.0473 | 0.0772 | 0.0479 | 0.0628 | 0.0764 | 0.1146 | 0.0959 | 0.1163 | 0.1301 | 0.1962 | 0.1359 | 0.1687 |
| | + LGHRec | 0.0495 | 0.0800 | 0.0499 | 0.0646 | 0.0797 | 0.1192 | 0.1030 | 0.1206 | 0.1350 | 0.2048 | 0.1427 | 0.1746 |
| | RelaImpr ↑ | 4.69% | 3.66% | 4.23% | 2.81% | 4.28% | 4.02% | **7.36%** | 3.68% | 3.79% | 4.38% | 5.01% | 3.49% |
| LightCCF | Base | 0.0485 | 0.0802 | 0.0484 | 0.0644 | 0.0347 | 0.0577 | 0.0375 | 0.0462 | 0.1314 | 0.1996 | 0.1402 | 0.1748 |
| | + LGHRec | 0.0521 | 0.0826 | 0.0504 | 0.0674 | 0.0370 | 0.0599 | 0.0389 | 0.0490 | 0.1412 | 0.2052 | 0.1488 | 0.1835 |
| | RelaImpr ↑ | **7.33%** | 3.03% | 4.12% | 4.70% | **6.71%** | 3.78% | 3.63% | 6.13% | 7.46% | 2.81% | 6.11% | 4.97% |
| TALLRec | Base | 0.0461 | 0.0761 | 0.0449 | 0.0625 | 0.0331 | 0.0548 | 0.0369 | 0.0452 | 0.1334 | 0.1934 | 0.1410 | 0.1734 |
| | LGHRec(LightCCF) | 0.0521 | 0.0826 | 0.0504 | 0.0674 | 0.0370 | 0.0599 | 0.0389 | 0.0490 | 0.1412 | 0.2052 | 0.1488 | 0.1835 |
| | RelaImpr ↑ | 13.02% | 8.54% | 12.25% | 7.84% | 11.78% | 9.31% | 5.42% | 8.41% | 5.84% | 6.10% | 5.53% | 5.82% |
| SPRec | Base | 0.0464 | 0.0778 | 0.0465 | 0.0631 | 0.0338 | 0.0556 | 0.0362 | 0.0439 | 0.1322 | 0.1921 | 0.1379 | 0.1720 |
| | LGHRec(LightCCF) | 0.0521 | 0.0826 | 0.0504 | 0.0674 | 0.0370 | 0.0599 | 0.0389 | 0.0490 | 0.1412 | 0.2052 | 0.1488 | 0.1835 |
| | RelaImpr ↑ | 12.28% | 6.17% | 8.39% | 6.81% | 9.47% | 7.73% | 7.46% | 11.62% | 6.80% | 6.81% | 7.90% | 6.68% |

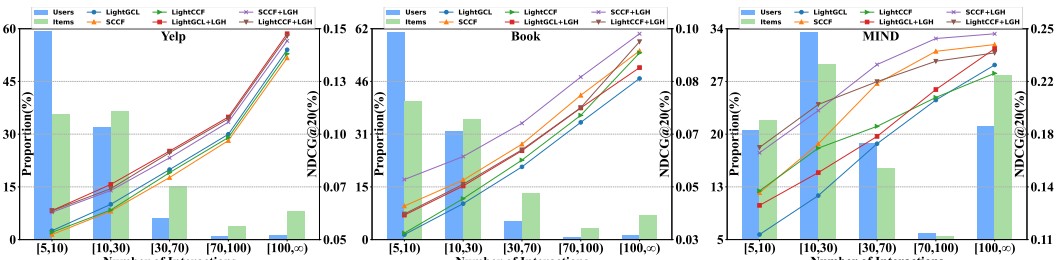

Figure 5: A comparison of NDCG@20 between LGHRec and baseline models across three datasets, grouped by different user and item interaction levels based on interaction count.

using only ID embeddings (a) form dispersed and unevenly dense clusters, making it difficult to distinguish semantically similar items. After introducing CoT semantic information from LLMs (b), the embedding distribution becomes more coherent, improving the discriminability of the embeddings. Embeddings learned by the full LGHRec model (c) exhibit a more uniform and dispersed distribution, indicating that the model is able to learn finer features to better distinguish different items.

**Adaptive Temperature Coefficient.** To verify HGPO can select the optimal temperature coefficient $\tau$ for different degrees nodes, we visualized the average $\tau$ selected by HGPO on the Yelp dataset. As shown in Figure 7(a), HGPO assigns larger $\tau$ values to low degree nodes, with $\tau$ gradually decreasing as node degree increases. This adaptive behavior demonstrates the effectiveness of HGPO: smaller $\tau$ values enhance the feature discriminability of high activity nodes, while larger $\tau$ values stabilize the learning process for low activity nodes in sparse data scenarios. In this way, HGPO dynamically adjusts the strength of contrastive learning based on node characteristics.

**Negative Sampling Analysis.** We compared similarity distribution for negative samples selected by HGPO and random sample on Yelp. As shown in Figure 7(b), 39.83% of the negative samples selected by HGPO fall within the hard negative range $[0.5, 0.8]$, which is much higher than the 20.39% through random sample. This increase is due to $R_{hard}$ incentivize the selection of rich information samples. In contrast, only 15.37% of easy negative samples (similarity $< 0.2$) were selected by HGPO, compared to 45.67% from random sample, as $R_{false}$ penalizes easy negatives. Although HGPO selected more false negative samples (9.51% versus 4.32% with random sampling), this reflects HGPO's exploration of the boundaries of hard samples. With $R_{easy}$ controlling the selection of false negatives, HGPO balances exploration and risk, focusing on rich information hard negatives.

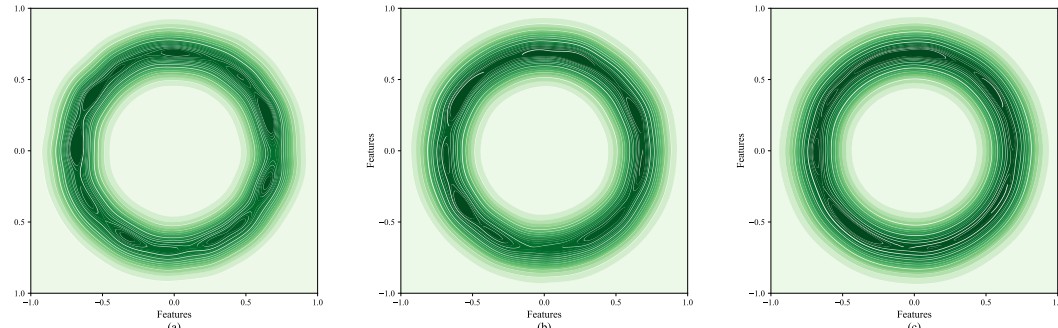

Figure 6: The KDE visualizes the distribution of item embeddings.

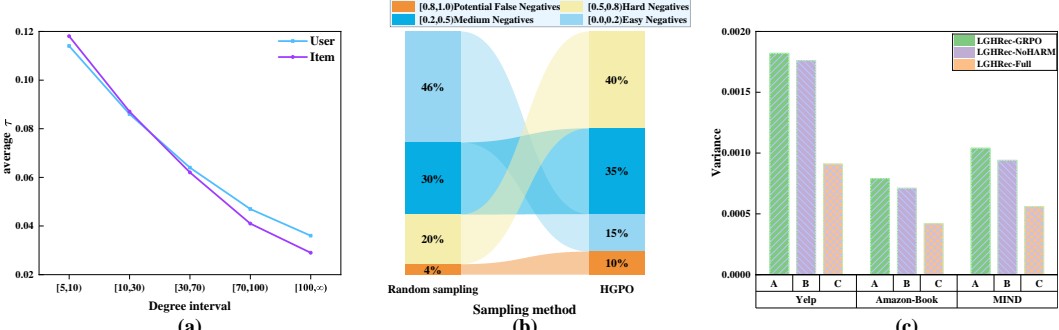

Figure 7: The results of various mechanisms of HGPO on the Yelp dataset.

**Effect of Coordination Mechanism.** The coordination loss $L^{HARM}$ addresses strategy inconsistency that arises when GRPO is applied to different activity groups, which can result in over optimization of certain groups at the expense of others. We compared NDCG@20 variance across user activity groups for the full LGHRec, LGHRec-NoHARM (without coordination loss), and LGHRec-GRPO (with HGPO replaced by GRPO). As shown in Figure 7(c), the full LGHRec exhibited the lowest performance variance across activity groups. When coordination loss was removed, the performance variance of LGHRec-NoHARM increased and became similar to LGHRec-GRPO. This result indicates that the coordination loss helps align strategies across groups by penalizing differences in average rewards. It prevents the over optimization of high activity groups.

### 3.3 HYPERPARAMETER SENSITIVITY

We use LightCCF as backbone and adjust the coordination weight ($\lambda_{harm}$), entropy coefficient ($c_1$), and temperature reward coefficient ($w_5$) on three datasets. We observe the NDCG@20, as shown in Figure 8. When the coordination weight $\lambda_{harm} = 0$, the model degenerates into GRPO, result in low performance and confirm the effectiveness of the coordination mechanism. As $\lambda_{harm}$ increases to around 0.5, performance improves due to better alignment of strategies across different groups. However, if $\lambda_{harm}$ becomes too high, performance slightly decreases because of excessive emphasis on consistency. For the entropy coefficient $c_1$, setting it to 0 leads to insufficient exploration and suboptimal performance. A small value promotes exploration and improves performance, while a large value causes the strategy to become too random, result in performance degradation. Regard the temperature reward coefficient $w_5$, when set it to 0 causes a performance drop. Increasing $w_5$ encourages the model to learn temperature adjustment strategies for heterogeneous data, with performance peaking at $w_5 = 1.2$. However, if $w_5$ is too high, the model focuses excessively on the temperature reward and neglects negative sample selection, thereby reducing performance.

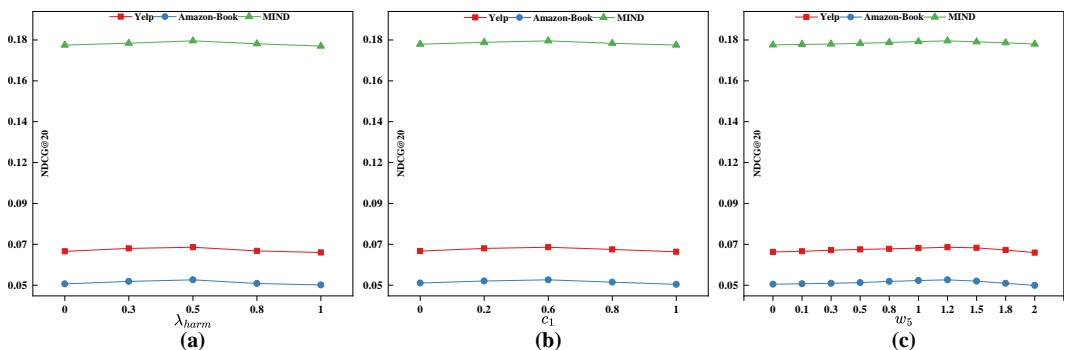

Figure 8: The results of three representative hyperparameter experiments.

Table 2: Results of ablation experiments performed on three datasets

| Variants | Yelp2018 | Amazon-Book | MIND | Explanation |
|---|---|---|---|---|
| LGHRec (Full) | 0.00% | 0.00% | 0.00% | Full model with all components. |
| **Impact of DSEG:** | | | | |
| LGHRec-NoDSEG | -6.12% | -7.96% | -5.26% | Without semantic enhancement, using LightCCF+HGPO. |
| LGHRec-RawText | -3.53% | -4.82% | -3.31% | Using raw text embeddings without CoT. |
| LGHRec (Basic LLM CoT) | -2.58% | -3.14% | -2.10% | Using LLM-generated CoT embeddings without fine-tuning. |
| LGHRec (Domain-tuned CoT) | -1.83% | -2.32% | -1.32% | Using LLM-generated CoT embeddings with domain fine-tuning. |
| LGHRec (CoT Task Fine-tuned) | -0.91% | -1.21% | -0.61% | Using LLM-generated CoT embeddings with CoT task fine-tuning. |
| LGHRec (Weighted Sum Fusion) | -1.10% | -1.52% | -0.88% | Using weighted summation for feature fusion. |
| **Impact of HGPO:** | | | | |
| LGHRec-NoHGPO | -4.27% | -5.23% | -3.98% | Use standard contrastive loss (fixed $\tau$, random negative sampling) |
| LGHRec-NoAdaptiveTau | -2.12% | -2.72% | -2.13% | HGPO optimizes only negative sampling with $\tau$ fixed |
| LGHRec-RandomNeg | -2.83% | -3.41% | -2.79% | HGPO only optimizes $\tau$, using random negative sampling |
| LGHRec-GRPO | -1.34% | -1.71% | -1.14% | Using GRPO optimization without the coordination mechanism |

## 3.4 ABLATION EXPERIMENT

We evaluated NDCG@20 on three datasets, the results in Table 2. The full LGHRec outperformed LGHRec-NoDSEG, demonstrate that LLM-generated semantic IDs enhance the information density and quality of the graph model's representations. The CoT reasoning capability of LLMs proved crucial, because LGHRec provide higher quality representations compare to LGHRec-RawText, which uses only raw text embeddings. The mixed fine-tuning method achieve the best results, effectively generate high quality semantic IDs, balance recommendation performance and general capabilities. In feature fusion, concatenation and linear layer outperformed weighted summation. Regard the impact of HGPO, the full LGHRec outperformed LGHRec-NoHGPO, confirm the effectiveness of HGPO. LGHRec also performed better than LGHRec-GRPO, highlight the importance of the coordination loss in HGPO for ensuring consistency across groups. Remove the adaptive temperature adjustment mechanism in LGHRec-NoAdaptiveTau led to performance drop, indicate that adaptive temperature adjustment is essential. Finally, LGHRec-RandomNeg, which uses random negative sampling, performed worse than the full LGHRec, demonstrate the superiority of guiding the agent to select rich information negative samples through reinforcement learning.

## 4 CONCLUSION

We propose LGHRec, which leverages the CoT reasoning ability of LLMs to generate semantic IDs for items offline, thereby guiding the collaborative filtering process of GNNs. LGHRec employs HGPO to optimize graph contrastive learning through strategic negative sampling, cross-group coordination, and adaptive temperature adjustment. The main contribution of LGHRec is its ability to enhance representation quality and information density via CoT reasoning, while optimizing contrastive learning with HGPO. This approach balances semantic richness, efficiency, and data heterogeneity. Experimental results show that LGHRec outperforms several graph contrastive learning models, demonstrating the effectiveness of combining LLM CoT reasoning with reinforcement learning in graph recommender systems. Future research will explore integrating multimodal information, such as images, into LLM CoT reasoning to generate item representations that incorporate both visual and textual semantic understanding.

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

**Appendix**

# A RELATED WORK

## A.1 LLM RECOMMENDATION

Research on using LLMs for recommendations can be categorized into two types. The first category is **LLM-as-RSs**, where LLMs are directly employed as recommender systems Bao et al. (2023); Zhang et al. (2024d); Luo et al. (2024); Lei et al. (2024); Zhang et al. (2024c). These systems are fine-tuned to generate recommendation lists. For instance, RecGPT Zhang et al. (2024d) fine-tunes a Transformer model for sequential recommendation; EITGR Chen et al. (2024) uses LLMs to generate item labels and adapt tokenization strategies dynamically; SPRec Gao et al. (2025) generates positive and negative samples iteratively to optimize preference alignment; TALLRec Bao et al. (2023) optimizes through instruction fine-tuned and recommendation task refinement; and RLRF4Rec Sun et al. (2024) generates user preferences with LLMs and refines them through feedback training. Although this approach offers end-to-end generative capabilities, it faces challenges such as low efficiency, underuse of collaborative signals, and hallucination problems. The second category is **LLM-enhanced RSs**, which improve existing recommendation models using the output text from LLMs. Pretrained language models Devlin et al. (2019) or specialized embedding methods Xiao et al. (2024) extract embeddings from LLM-generated text, which are then used as item features in recommendation models Xi et al. (2024); Zhang et al. (2024a); Liu et al. (2024b); Qu et al. (2024); Deng et al. (2025). For example, KAR Xi et al. (2024) integrates knowledge extracted from LLMs into the model; Du et al. (2024) optimizes representations by extracting explicit and implicit user information; and EmbSum Zhang et al. (2024a) uses LLMs to generate user interests as model features. This approach is more flexible and friendly, but current methods have not fully leveraged the chain-of-thought (CoT) reasoning capabilities of LLMs to produce higher quality, more rich information semantic representations.

## A.2 GNN-BASED REC

Some methods He et al. (2020) rely on ID features and make insufficient use of textual semantic information. To address this, some studies attempt to combine large language models (LLMs) with graph neural networks (GNNs) to enhance graph structures or node features. For instance, LLMs are used to infer potential relationships between nodes and add edges Zhao et al. (2024); Liu et al. (2025); Zhang et al. (2024b). One approach, LLM-KERec Zhao et al. (2024), extracts entities and infers relationships using LLMs to construct supplementary graphs. Another method introduces latent factors inferred by LLMs as new nodes to build richer semantic graphs Shan et al. (2024); Jeon et al. (2024); Hu et al. (2025), such as AutoGraph Shan et al. (2024), which uses LLMs to infer user preferences and item knowledge, encoding them into semantic vectors and extracting latent factors as graph nodes. Other studies have enhanced GNN node feature representations directly with LLMs Sakurai et al. (2025), for example, by extracting preference features to drive reinforcement learning Sakurai et al. (2025). Some works Li et al. (2024); Chen & Suzumura (2024) design specific prompts to input node text into LLMs for enhancement, then encode the output as embeddings. Our LGHRec also follows the node feature enhancement paradigm but distinguishes itself by utilizing the chain-of-thought (CoT) reasoning capability of LLMs. Instead of generating simple text descriptions, LGHRec produces text with deeper logic and analysis, aiming to obtain semantic IDs with higher information density.

## A.3 GRAPH CONTRASTIVE LEARNING

Graph contrastive learning is used in recommender systems to address the issue of data sparsity Lin et al. (2022); Wu et al. (2021); Yang et al. (2023); Zhang et al. (2025b); Sun et al. (2023); Yu et al. (2022); Wu et al. (2024b); Cai et al. (2023). Some methods include SGL Wu et al. (2021) (data augmentation), SimGCL Yu et al. (2022) (add noise), LightGCL Cai et al. (2023) (SVD enhancement), and NCL Lin et al. (2022) (structural and semantic prototype contrast). However, these approaches face challenges such as false negative samples and fixed temperature coefficients. Additionally, when using the GRPO reinforcement learning algorithm to handle groups with varying node degrees, it fails to ensure coordination between the strategies of different groups. This leads to

Table 3: Basic statistics of datasets.

| Datasets | Users | Items | Interactions | Density |
|---|---|---|---|---|
| Yelp2018 | 277631 | 112394 | 4250483 | 0.00013 |
| Amazon-Book | 1856344 | 704093 | 27164983 | 0.00002 |
| MIND | 47495 | 28420 | 5311336 | 0.00393 |

optimization biased toward active users and popular items, causing performance imbalances. The HGPO algorithm we propose helps mitigate this issue.

# B EXPERIMENTAL SETUP

## B.1 DATASETS

We conducted experiments on three datasets: Yelp2018[1], Amazon-Book[2], and MIND[3]. For each dataset, we applied a 5-core setting, randomly selecting 80% of the data for training, 10% for validation, and the remaining 10% for testing. Detailed statistics for the datasets are provided in Table 3. The extremely low density of these datasets underscores the issue of data sparsity, which is the primary challenge addressed by both graph contrastive learning and our proposed HGPO algorithm.

## B.2 BASELINES

To evaluate the effectiveness of LGHRec, we selected the following models as backbones and integrated the DSEG and HGPO components of LGHRec into them for comparison:

- **SGL** Wu et al. (2021): Generates node views through node deletion, edge deletion, and random walks.

- **SimGCL** Yu et al. (2022): Introduces graph contrastive learning by omitting explicit graph augmentation.

- **LightGCL** Cai et al. (2023): Utilizes singular value decomposition for graph augmentation in contrastive learning.

- **VGCL** Yang et al. (2023): Proposes a variational graph generation contrastive learning framework, generating contrastive views through sampling.

- **NESCL** Sun et al. (2023): Introduces neighborhood-enhanced contrastive learning, treating the anchor's collaborative neighbors as positive samples.

- **SCCF** Wu et al. (2024b): Proposes contrastive collaborative filtering, capturing higher-order connectivity based on an improved contrastive loss function, without requiring graph convolution layers.

- **CIKG** Hu et al. (2025): It uses LLMs to infer user interests, structuring this knowledge into a hybrid graph for enhanced recommendation.

- **AutoGraph** Shan et al. (2024): It leverages LLMs to create semantic vectors and uses vector quantization to extract latent factors for graph construction.

- **LightCCF** Zhang et al. (2025b): Introduces lightweight contrastive learning by incorporating neighborhood aggregation objectives.

- **TALLRec** Bao et al. (2023): It aligns LLM with recommendation tasks by formatting data as instructions and utilizing an two-stage LoRA-based fine-tuning process.

- **SPRec** Gao et al. (2025): It addresses biases that emerge during LLM fine-tuning by employing Direct Preference Optimization.

---

[1]https://business.yelp.com/data/resources/open-dataset/
[2]http://jmcauley.ucsd.edu/data/amazon/
[3]https://msnews.github.io/

Table 4: Computational cost and efficiency of LGHRec.

| Dataset | Model Variant | Pre-processing LLM CoT Generate(minute) | Pre-processing BERT Encoding(minute) | Pre-processing Complexity | Train (Sec/Epoch) | Train Complexity | Train Memory | Train Total Time | Inference (Ms/Batch) |
|---|---|---|---|---|---|---|---|---|---|
| Yelp2018 | Baseline(LightCCF) | N/A | N/A | N/A | $11.8 \pm 0.16$ | $O(L \cdot |E| \cdot d)$ | 3.26GB | 108.32m | $25.3 \pm 0.7$ |
| | Baseline+DSEG | 149.3 | 2.3 | $O(|I|)$ | $12.4 \pm 0.13$ | $O(L \cdot |E| \cdot d)$ | 3.31GB | 84.92m | $28.9 \pm 0.6$ |
| | + LGHRec(Full Model) | 149.3 | 2.3 | $O(|I|)$ | $16.1 \pm 0.84$ | $O(L \cdot |E| \cdot d + |V|)$ | 3.53GB | 150.28m | $29.8 \pm 0.8$ |
| Amazon-Book | Baseline(LightCCF) | N/A | N/A | N/A | $46.8 \pm 4.35$ | $O(L \cdot |E| \cdot d)$ | 3.38GB | 400.92m | $25.6 \pm 0.9$ |
| | Baseline+DSEG | 903.7 | 15.2 | $O(|I|)$ | $48.6 \pm 4.98$ | $O(L \cdot |E| \cdot d)$ | 3.49GB | 318.74m | $29.3 \pm 0.6$ |
| | + LGHRec(Full Model) | 903.7 | 15.2 | $O(|I|)$ | $59.9 \pm 3.25$ | $O(L \cdot |E| \cdot d + |V|)$ | 3.85GB | 563.28m | $29.9 \pm 0.8$ |
| Steam | Baseline(LightCCF) | N/A | N/A | N/A | $10.5 \pm 0.13$ | $O(L \cdot |E| \cdot d)$ | 3.23GB | 81.34m | $23.1 \pm 0.5$ |
| | Baseline+DSEG | 37.9 | 0.6 | $O(|I|)$ | $11.2 \pm 0.19$ | $O(L \cdot |E| \cdot d)$ | 3.27GB | 69.45m | $27.9 \pm 0.5$ |
| | + LGHRec(Full Model) | 37.9 | 0.6 | $O(|I|)$ | $15.6 \pm 0.49$ | $O(L \cdot |E| \cdot d + |V|)$ | 3.39GB | 124.68m | $28.5 \pm 0.6$ |

## B.3 EVALUATION METRICS

We used two Top-K metrics He et al. (2020); Wu et al. (2021): Recall@K and NDCG@K, and report results for $K = 10$ and $K = 20$. The all-ranking evaluation strategy Zhao et al. (2020) was adopted, where, for each user in the test set, the model is required to predict and rank the scores of all items that the user has not interacted with. The hit rate of the top-K ranked items is then evaluated.

## B.4 IMPLEMENTATION DETAILS

All methods use the Adam optimizer with a learning rate of 0.001, a batch size of 4096, and an embedding dimension of 64. The GNN consists of three layers. We employ an early stopping strategy with a patience value of 10 to prevent overfitting. The number of negative samples, $K$, is set to 10% of the total number of nodes. The reward function thresholds are: $\theta_{FN} = 0.8$, $\theta_{easy} = 0.5$, $\theta_{FP} = 0.8$, and $\theta_{easy\_low} = 0.2$. The clipping range $\varepsilon$ for HGPO is 0.2. The learning rate for the policy network is 0.0001. The entropy regularization coefficient is $c_1 \in \{0, 0.2, 0.6, 0.8, 1.0\}$, the coordination loss coefficient is $\lambda_{harm} \in \{0, 0.3, 0.5, 0.8, 1.0\}$, and the temperature reward coefficient $w_5 \in \{0, 0.1, 0.3, 0.5, 0.8, 1.0, 1.2, 1.5, 1.8, 2.0\}$. Users and items are divided into five groups based on their degree. We use the Qwen2.5-32B-Instruct model, fine-tuned with a mixed fine-tuning strategy, to generate CoT text for the items. Pre-trained BERT is used to extract embeddings. All experiments are implemented in PyTorch on a server equipped with eight NVIDIA A100 GPUs.

## B.5 COMPUTATIONAL COST AND EFFICIENCY

Our approach maintains efficient online inference by shifting the costly LLM computations to a one-time offline preprocessing stage. We analyzed the computational cost in detail. The results are shown in Table 4. Here, N/A indicates that the baseline lacks an LLM inference stage, N is the number of items, V is the total number of nodes, E is the total number of interactions, L is the number of GNN layers, and d is the embedding dimension. All measurements were performed on a single NVIDIA A100 GPU with a batch size of 4096. By front-loading the expensive CoT inference into the offline phase, our method preserves high efficiency during online serving.

The computational cost of LGHRec is divided into two parts. First, there is the one-time offline preprocessing cost. The DSEG module uses an LLM to generate CoT semantic ID vectors, with a time complexity of O(I). On a large dataset like Amazon-Book, this step takes about 15 hours. Although substantial, this one-time cost is acceptable for industrial applications. The resulting high-quality semantic IDs can be stored and reused indefinitely. Additionally, inference time on large datasets can be further reduced by leveraging distributed acceleration frameworks such as vLLM. Second, the training-phase cost increases with the HGPO module. For example, on Yelp, each epoch's training time rises from 11.8 s to 16.1 s. We consider this a reasonable trade-off given the model's performance gains.

The key advantage of our approach lies in the guarantee of online inference efficiency. As shown in the Inference column of Table 4, LGHRec's per-batch latency remains nearly unchanged compared to the baselines. For example, on Amazon-Book, latency increases only from 25.6ms to 29.9ms. In an online recommendation service, the system only invokes the pre-trained GNN model to compute user and item embeddings. It does not perform costly LLM inference or run the HGPO policy network. This decoupled design delivers significant accuracy gains while meeting industry requirements for low latency and high throughput. These results demonstrate LGHRec's practicality for real-world deployment.

Table 5: Impact of different LLM architectures on the performance of LGHRec.

| Variants | Yelp2018 | | Amazon-Book | | MIND | |
|---|---|---|---|---|---|---|
| | Recall@10 | NDCG@10 | Recall@10 | NDCG@10 | Recall@10 | NDCG@10 |
| Qwen-2.5-32B-Instruct(Base) | — | — | — | — | — | — |
| DeepSeek-R1-Distill-Qwen-14B | -0.96% | -0.77% | -0.81% | -0.77% | -0.72% | -0.77% |
| Llama-3.1-8B-Instruct | -1.34% | -1.59% | -1.35% | -1.03% | -0.93% | -0.99% |
| Qwen-2.5-7B-Instruct | -1.15% | -1.19% | -1.08% | -0.77% | -0.80% | -0.87% |
| Qwen3-8B | -0.77% | -0.99% | -0.54% | -0.51% | -0.64% | -0.78% |
| Qwen3-32B | **+0.96%** | **+1.19%** | **+1.08%** | **+1.03%** | **+0.83%** | **+0.98%** |

## B.6 IMPACT OF DIFFERENT LLM ARCHITECTURES ON THE PERFORMANCE OF LGHREC.

We evaluated multiple LLM architectures, with results summarized in Table 5. Qwen-2.5-32B-Instruct served as our baseline. These findings support two key conclusions. First, Within a single model family, performance increases with model size. For instance, Qwen3-32B outperforms its smaller variants. Its superior context comprehension and reasoning capabilities generate higher-quality Chain-of-Thought outputs. Second, When comparing across model families, we find that the performance of LGHRec is correlated with the general ability of LLM, such as MMLU score, and the models with stronger general ability improve the recommendation performance more, such as the newer Qwen3 series models, which is consistent with the conclusions of related studies **??**. We demonstrate that our framework effectively leverages and benefits from more advanced LLMS.

## B.7 SENSITIVITY ANALYSIS ON THE FALSE NEGATIVE THRESHOLD

We conducted experiments on two representative datasets, Yelp2018 and Amazon-Book. We performed a hyperparameter sensitivity analysis on the four similarity thresholds in the reward function $\theta_{\mathrm{FN}}$, $\theta_{\mathrm{easy}}$, $\theta_{\mathrm{FP}}$ and $\theta_{easy\_low}$ . We used LightCCF, a strong performer in our paper, as the backbone network. For each analysis, we varied one threshold within a reasonable range while keeping all other parameters at their optimal settings as reported in the paper, and we observed the changes in the NDCG@20 metric. The results summarized in Table 6.

$\theta_{FN}$ defines the similarity lower bound for the false negatives, and indirectly sets the similarity upper bound for the hard negatives $[\theta_{\mathrm{easy}}, \theta_{\mathrm{FN}})$. An appropriate $\theta_{FN}$ is crucial for the model to distinguish between false and true negatives. We tested its value within the range $[0.7, 0.9]$. From the results, we observe that the model achieves its best performance when $\theta_{FN}$ is set to 0.8. A too low $\theta_{FN}$ (e.g., 0.7) narrows the valid range and may mistakenly penalize some informative hard negatives as false negatives, thereby impairing learning. Conversely, a too high $\theta_{FN}$ (e.g., 0.9) relaxes the judgment of false negatives, potentially preventing the model from effectively identifying and excluding samples that are truly similar to the anchor, which also degrades performance. Overall, the model maintains high stability in the range $[0.75, 0.85]$.

$\theta_{\mathrm{easy}}$ defines the similarity lower bound for the hard negatives, used to distinguish between hard negatives and easy negatives. It determines the difficulty range of negative samples that the model focuses on. We tested its value within the range $[0.3, 0.7]$. When $\theta_{\mathrm{easy}}$ is set to 0.5, the model can effectively identify and reward the most informative hard negatives. If this value is too low (e.g., 0.3), many low discrimination easy negatives are mistakenly treated as hard negatives, reducing the value of the reward signal. Conversely, if it is too high (e.g., 0.7), the range of hard negatives becomes overly narrow, causing the model to miss many valuable training signals and leading to a more pronounced performance drop. Experimental results indicate that the model's performance remains relatively stable within the range $[0.4, 0.6]$.

$\theta_{FP}$ is used to penalize negative samples that are overly similar to the positive sample rather than to the anchor user, helping to avoid selecting negatives that are semantically very close to the positive. We tested its value within the range $[0.7, 0.9]$. The experiments show that $\theta_{FP}$ performs best at 0.8. A too low $\theta_{FP}$ (e.g., 0.7) makes the model overly conservative, mistakenly penalizing some high quality hard negatives that only have moderate similarity to the positive sample. Conversely, a too high $\theta_{FP}$ (e.g., 0.9) fails to effectively exclude negatives that are highly similar to the positive sample and may cause confusion. The sensitivity of this parameter is relatively low, with performance fluctuations minor within the range $[0.75, 0.85]$, demonstrating good robustness.

$\theta_{\text{easy\_low}}$ penalizes samples whose similarity falls below this threshold, preventing the model from wasting learning opportunities on overly simple negatives. We tested its value within the range $[0.1, 0.3]$. When $\theta_{\text{easy\_low}}$ is set to 0.2, the model achieves optimal performance. This indicates that moderately penalizing the least informative negatives is beneficial. If the value is too low (e.g., 0.1), the penalty on easy negatives is insufficient, and the model may still select ineffective samples. Conversely, if it is too high (e.g., 0.3), it may mistakenly penalize samples that carry weak but useful signals, slightly harming performance. The parameter remains stable within the range $[0.15, 0.25]$.

From the above experiments, we observe that although the choice of these similarity thresholds does affect final performance, LGHRec maintains strong stability within reasonable variations of these values. This demonstrates the robustness of our method and confirms that the default values chosen in our paper are empirically justified.

Table 6: Hyperparameter sensitivity analysis for reward function thresholds on Yelp2018 and Amazon-Book datasets, evaluated using NDCG@20.

| Parameter | Value | Yelp2018 (NDCG@20) | Amazon-Book (NDCG@20) |
|---|---|---|---|
| $\theta_{easy}$ | 0.3 | 0.0658(-2.30%) | 0.0478(-2.45%) |
| | 0.4 | 0.0667(-0.96%) | 0.0485(-1.02%) |
| | **0.5 (Best)** | **0.0674(0.00%)** | **0.0490(0.00%)** |
| | 0.6 | 0.0664(-1.34%) | 0.0483(-1.43%) |
| | 0.7 | 0.0653(-3.07%) | 0.0475(-3.06%) |
| $\theta_{easy\_low}$ | 0.10 | 0.0663(-1.54%) | 0.0483(-1.43%) |
| | 0.15 | 0.0670(-0.58%) | 0.0487(-0.61%) |
| | **0.20 (Best)** | **0.0674(0.00%)** | **0.0490(0.00%)** |
| | 0.25 | 0.0668(-0.77%) | 0.0486(-0.82%) |
| | 0.30 | 0.0661(-1.92%) | 0.0481(-1.84%) |
| $\theta_{FN}$ | 0.70 | 0.0661(-1.92%) | 0.0481(-1.84%) |
| | 0.75 | 0.0668(-0.77%) | 0.0486(-0.82%) |
| | **0.80 (Best)** | **0.0674(0.00%)** | **0.0490(0.00%)** |
| | 0.85 | 0.0670(-0.58%) | 0.0487(-0.61%) |
| | 0.90 | 0.0662(-1.73%) | 0.0482(-1.63%) |
| $\theta_{FP}$ | 0.70 | 0.0664(-1.34%) | 0.0484(-1.22%) |
| | 0.75 | 0.0670(-0.58%) | 0.0488(-0.41%) |
| | **0.80 (Best)** | **0.0674(0.00%)** | **0.0490(0.00%)** |
| | 0.85 | 0.0671(-0.38%) | 0.0489(-0.20%) |
| | 0.90 | 0.0666(-1.15%) | 0.0486(-0.82%) |

## C  CONVERGENCE ANALYSIS OF THE HGPO OBJECTIVE FUNCTION

### C.1  REVIEW OF THE HGPO ALGORITHM

The HGPO algorithm's objective function $L_{HGPO}(\theta)$ is defined as:

$$L_{HGPO}(\theta) = -L^{\text{POLICY}}(\theta) + c_1 S[\pi_\theta] + L^{\text{HARM}}(\theta) \quad \text{(Eq. A)} \tag{14}$$

where:

1. $L^{\text{POLICY}}(\theta)$ is the policy loss based on the relative advantage function with clipping:

$$L^{\text{POLICY}}(\theta) = \widehat{\mathbb{E}}_t \Big[ \min\big(r_t(\theta) A_t^{\text{rel}}, \text{clip}(r_t(\theta), 1-\epsilon, 1+\epsilon) A_t^{\text{rel}}\big) \Big] \quad \text{(Eq. B)} \tag{15}$$

   Here, $r_t(\theta) = \frac{\pi_\theta(a_t|s_t)}{\pi_{\theta_{\text{old}}}(a_t|s_t)}$ is the importance-sampling weight, and the relative advantage is as follow:

$$A_t^{\text{rel}} = r_t - \bar{R}_{g(s_t)} \tag{16}$$

   where $r_t$ is the immediate reward for taking action $a_t$ in state $s_t$, and $\bar{R}_{g(s_t)}$ is the average reward of the group $g$ containing state $s_t$.

2. $S[\pi_\theta]$ is the entropy regularization term of the policy:

$$S[\pi_\theta] = \widehat{\mathbb{E}}_t\big[H\big(\pi_\theta(\cdot \mid s_t)\big)\big] \quad \text{(Eq. C)} \tag{17}$$

with

$$H\big(\pi_\theta(\cdot \mid s_t)\big) = H_{\text{neg}}\big(\pi_\theta(a_{\text{neg}} \mid s_t)\big) + H_{\text{temp}}\big(\pi_\theta(a_{\text{temp}} \mid s_t)\big) \tag{18}$$

3. $L^{\text{HARM}}(\theta)$ is the harmonization loss, which penalizes variance in average rewards across groups:

$$L^{\text{HARM}}(\theta) = \lambda_{\text{harm}} \, \text{Var}_{g\in\mathcal{G}}\big[\bar{R}_g\big] \quad \text{(Eq. D)} \tag{19}$$

We now proceed to prove that, when optimized via gradient descent, the HGPO algorithm converges to a (local) minimum of $L_{HGPO}(\theta)$.

## C.2 Proof Outline

We follow the standard convergence proof approach for optimization algorithms:

1. **Boundedness**: Show that the objective function $L_{HGPO}(\theta)$ is lower-bounded under appropriate conditions.

2. **Sufficient Decrease**: Prove that at each iteration, if the gradient is non-zero, the objective value decreases sufficiently (or at least does not increase).

3. **Convergence to a Stationary Point**: By combining boundedness and sufficient decrease, demonstrate that the gradient of the policy parameters $\theta$ converges to zero, meaning the algorithm converges to a stationary point.

## C.3 Detailed Mathematical Proof

### C.3.1 Preliminaries

**Boundedness of the Reward.** The reward at time step $t$ consists of a rule-based component $R_t$ (composed of $R_{\text{hard}}, R_{\text{false}}, R_{\text{easy}}$ as defined in Eqs. 1, 2, 3) and an adaptive temperature component $R_\tau$ (Eq. 5). We show that both components are bounded.

First, each of $R_{\text{hard}}, R_{\text{false}}, R_{\text{easy}}$ is defined using similarity thresholds. Since cosine similarity lies in the range $[-1, 1]$ and the weights $w_1, w_2, w_3, w_4$ are fixed constants, these terms are bounded.

Second, the adaptive temperature reward is defined as follows:

$$R_\tau = -w_5\big|\tau_u^{(t)} - T_{\text{ideal}}(d_u)\big|, \quad T_{\text{ideal}}(d_u) = \frac{1}{1 + \log(1 + d_u)} \tag{20}$$

For a finite node degree $d_u$, $T_{\text{ideal}}(d_u)$ is bounded. The action space $\tau_u^{(t)}$ also lies within a bounded interval $[\tau_{\text{min}}, \tau_{\text{max}}]$, where $\tau_{\text{min}} > 0$. Therefore, $R_\tau$ is bounded.

As a result, the total reward $r_t = R_t + R_\tau$ is bounded. In other words, there exist constants $R_{\text{min}}$ and $R_{\text{max}}$ such that $R_{\text{min}} \le r_t \le R_{\text{max}}$.

**Smoothness of the Policy Function.** The policy network $\pi_\theta(a \mid s)$ is continuously differentiable with respect to its parameters $\theta$, and its gradient $\nabla_\theta\pi_\theta(a \mid s)$ is Lipschitz continuous. Additionally, both the action probability outputs and the network's parameter values are bounded.

**Boundedness of Importance Sampling Ratios.** The importance sampling ratio $r_t(\theta) = \frac{\pi_\theta(a_t|s_t)}{\pi_{\theta_{\text{old}}}(a_t|s_t)}$ is bounded in practice by applying clipping during updates, which prevents excessively large variance.

**Boundedness of Entropy.** For the discrete negative-sampling action $a_{\text{neg}}$, the entropy is given by:

$$H_{\text{neg}} = -\sum_{j=1}^{M_{\text{neg}}} p_j \log p_j, \quad 0 \le H_{\text{neg}} \le \log M_{\text{neg}} \tag{21}$$

where $M_{\text{neg}}$ is the size of the negative sample candidate pool.

For the continuous temperature selection action $a_{\text{temp}}$, under a Gaussian policy $\mathcal{N}(\mu_\theta(s_t), \sigma_\theta^2(s_t))$, the entropy is:

$$H_{\text{temp}} = \frac{1}{2} \log \left( 2\pi e \sigma_\theta^2(s_t) \right) \tag{22}$$

To ensure a lower bound and avoid degeneration as $\sigma \to 0$, we enforce $\sigma_\theta^2(s_t) \geq \sigma_{\min}^2 > 0$, which yields a positive lower bound for $H_{\text{temp}}$.

Therefore, the total entropy:

$$H\big(\pi_\theta(\cdot \mid s_t)\big) = H_{\text{neg}} + H_{\text{temp}} \tag{23}$$

is lower-bounded. Consequently, the entropy regularization term:

$$S[\pi_\theta] = \widehat{\mathbb{E}}_t \left[ H\big(\pi_\theta(\cdot \mid s_t)\big) \right] \tag{24}$$

is also bounded below.

### C.3.2 Proving the Boundedness of HGPO

We need to show that the objective function $L_{HGPO}(\theta)$ admits a finite lower bound. Recall that:

$$L_{HGPO}(\theta) = -L^{\text{POLICY}}(\theta) + c_1 S[\pi_\theta] + L^{\text{HARM}}(\theta) \tag{25}$$

**Analysis of the Policy Loss $L^{\text{POLICY}}(\theta)$.** The relative advantage is given by:

$$A_t^{\text{rel}} = r_t - \bar{R}_{g(s_t)} \tag{26}$$

Since the immediate reward $r_t$ is bounded, and the group-mean reward $\bar{R}_g = \mathbb{E}\big[r_t \mid s_t \in g\big]$ is also bounded, it follows that $A_t^{\text{rel}}$ is bounded. Let $|A_t^{\text{rel}}| \leq A_{\text{max\_abs}}$.

The importance-sampling weight $r_t(\theta)$ is strictly positive, i.e., $r_t(\theta) > 0$.

Next, we consider the clipped loss term $L_t^{\text{CLIP}}(\theta)$:

$$L_t^{\text{CLIP}}(\theta) = \min\Big( r_t(\theta) A_t^{\text{rel}}, \ \text{clip}\big(r_t(\theta), 1 - \epsilon, 1 + \epsilon\big) A_t^{\text{rel}} \Big) \tag{27}$$

- If $A_t^{\text{rel}} \geq 0$, the clipped loss term becomes:

$$L_t^{\text{CLIP}}(\theta) = \min\big( r_t(\theta) A_t^{\text{rel}}, \ (1 + \epsilon) A_t^{\text{rel}} \big) \tag{28}$$

  since $r_t(\theta)$ typically hovers around 1 and the upper clipping bound $(1+\epsilon)$ is active. Because $r_t(\theta) \geq 0$, it follows that $L_t^{\text{CLIP}}(\theta) \geq 0$. Moreover, since $r_t(\theta)$ is restricted to the interval $[0, C_{\text{ratio}}]$, we have: $L_t^{\text{CLIP}}(\theta) \leq C_{\text{ratio}} A_t^{\text{rel}}$.

- If $A_t^{\text{rel}} < 0$, the clipped loss term becomes:

$$L_t^{\text{CLIP}}(\theta) = \max\big( r_t(\theta) A_t^{\text{rel}}, \ (1 - \epsilon) A_t^{\text{rel}} \big) \tag{29}$$

  since $A_t^{\text{rel}}$ is negative, the minimum operation flips to a maximum, and the lower clipping bound $(1 - \epsilon)$ is active. Hence, $L_t^{\text{CLIP}}(\theta)$ is bounded. For example, if $r_t(\theta) \in [1 - \delta, 1 + \delta]$ (with $r_t(\theta)$ typically near 1), then:

$$L_t^{\text{CLIP}}(\theta) \in \big[ (1 - \epsilon) A_{\text{min\_neg}}, \ (1 + \epsilon) A_{\text{max\_pos}} \big] \tag{30}$$

  where $A_{\text{min\_neg}}$ is the lower bound of the negative advantages and $A_{\text{max\_pos}}$ is the upper bound of the positive advantages.

Therefore, the overall policy loss $L^{\text{POLICY}}(\theta) = \widehat{\mathbb{E}}_t\big[L_t^{\text{CLIP}}(\theta)\big]$ is bounded. Let $L_{\min}^{\text{POLICY}} \leq L^{\text{POLICY}}(\theta) \leq L_{\max}^{\text{POLICY}}$.

Hence, $-L^{\text{POLICY}}(\theta)$ is also bounded:

$$-L_{\max}^{\text{POLICY}} \leq -L^{\text{POLICY}}(\theta) \leq -L_{\min}^{\text{POLICY}} \tag{31}$$

**Analysis of the Entropy Regularization Term** $c_1 S[\pi_\theta]$. From the boundedness of entropy, the entropy regularization term $S[\pi_\theta]$ has a lower bound, denoted $S_{\min}$. If $c_1 > 0$ (as is typically chosen to encourage exploration), then: $c_1 S[\pi_\theta] \geq c_1 S_{\min}$.

**Analysis of the Harmonization Loss** $L^{\mathrm{HARM}}(\theta)$. The harmonization loss is defined as:

$$L^{\mathrm{HARM}}(\theta) = \lambda_{\mathrm{harm}} \operatorname{Var}_{g \in \mathcal{G}}\left[\bar{R}_g\right] \tag{32}$$

Since $\bar{R}_g$ is the mean reward of group $g$, and the immediate reward $r_t$ is bounded ($R_{\min} \leq r_t \leq R_{\max}$), we have: $R_{\min} \leq \bar{R}_g \leq R_{\max}$.

The variance $\operatorname{Var}_{g \in \mathcal{G}}\left[\bar{R}_g\right] = \mathbb{E}_g\left[\left(\bar{R}_g - \mathbb{E}_g[\bar{R}_g]\right)^2\right]$ of a bounded random variable is also bounded. In particular:

$$0 \leq \operatorname{Var}_{g \in \mathcal{G}}\left[\bar{R}_g\right] \leq \frac{(R_{\max} - R_{\min})^2}{4} \tag{33}$$

Since $\lambda_{\mathrm{harm}} \geq 0$, it follows that:

$$0 \leq L^{\mathrm{HARM}}(\theta) = \lambda_{\mathrm{harm}} \operatorname{Var}_g[\bar{R}_g] \leq \lambda_{\mathrm{harm}} \frac{(R_{\max} - R_{\min})^2}{4} \tag{34}$$

Thus, $L^{\mathrm{HARM}}(\theta)$ is both lower- and upper-bounded.

**Summary of the Lower Bound of** $L_{HGPO}(\theta)$. We now summarize the lower bound of $L_{HGPO}(\theta)$:

$$L_{HGPO}(\theta) \geq -L_{\max}^{\mathrm{POLICY}} + c_1 S_{\min} + \lambda_{\mathrm{harm}} \cdot 0 = -L_{\max}^{\mathrm{POLICY}} + c_1 S_{\min} \tag{35}$$

Therefore, $L_{HGPO}(\theta)$ is lower-bounded. We denote this bound by:

$$L_{HGPO,\min} = -L_{\max}^{\mathrm{POLICY}} + c_1 S_{\min} \tag{36}$$

### C.3.3 PROVING THE RELEVANCE OF THE GRADIENT

The HGPO algorithm updates the parameters $\theta$ via gradient descent:

$$\theta_{k+1} = \theta_k - \alpha_k \nabla_\theta L_{HGPO}(\theta_k) \tag{37}$$

where $\alpha_k$ is the learning rate.

Using the Taylor expansion for a differentiable function $f(x)$,

$$f(x') \approx f(x) + \nabla f(x)^T (x' - x) + \frac{1}{2}(x' - x)^T \mathbf{H}(x)(x' - x) \tag{38}$$

where $\mathbf{H}(x)$ is the Hessian matrix.

Applying this to $L_{HGPO}(\theta)$ at $\theta_k$, we get:

$$L_{HGPO}(\theta_{k+1}) \approx L_{HGPO}(\theta_k) + \nabla_\theta L_{HGPO}(\theta_k)^T (\theta_{k+1} - \theta_k) + \frac{1}{2}(\theta_{k+1} - \theta_k)^T \mathbf{H}_k(\theta_{k+1} - \theta_k) \tag{39}$$

Substituting $\theta_{k+1} - \theta_k = -\alpha_k \nabla_\theta L_{HGPO}(\theta_k)$, we get:

$$L_{HGPO}(\theta_{k+1}) - L_{HGPO}(\theta_k) \approx -\alpha_k \|\nabla_\theta L_{HGPO}(\theta_k)\|^2 + \frac{1}{2}\alpha_k^2 \nabla_\theta L_{HGPO}(\theta_k)^T \mathbf{H}_k \nabla_\theta L_{HGPO}(\theta_k) \tag{40}$$

Assuming $L_{HGPO}(\theta)$ is $L$-smooth, i.e., its gradient is Lipschitz continuous with constant $L$, the maximum eigenvalue of $\mathbf{H}_k$ satisfies $\lambda_{\max}(\mathbf{H}_k) \leq L$. Hence,

$$\nabla_\theta L_{HGPO}(\theta_k)^T \mathbf{H}_k \nabla_\theta L_{HGPO}(\theta_k) \leq L \|\nabla_\theta L_{HGPO}(\theta_k)\|^2 \tag{41}$$

Thus, we have:

$$L_{HGPO}(\theta_{k+1}) - L_{HGPO}(\theta_k) \leq -\alpha_k \|\nabla_\theta L_{HGPO}(\theta_k)\|^2 + \frac{L}{2}\alpha_k^2 \|\nabla_\theta L_{HGPO}(\theta_k)\|^2 \tag{42}$$

Simplifying, we get:

$$L_{HGPO}(\theta_{k+1}) - L_{HGPO}(\theta_k) \leq -\alpha_k \left(1 - \frac{L\alpha_k}{2}\right) \|\nabla_\theta L_{HGPO}(\theta_k)\|^2 \quad \text{(Eq. E)} \tag{43}$$

To guarantee a decrease in the objective, we require:

$$1 - \frac{L\alpha_k}{2} > 0 \quad \Longleftrightarrow \quad \alpha_k < \frac{2}{L} \tag{44}$$

If we choose a sufficiently small learning rate, e.g., $\alpha_k = \alpha < \frac{1}{L}$, then:

$$1 - \frac{L\alpha}{2} \geq \frac{1}{2} \tag{45}$$

Therefore:

$$L_{HGPO}(\theta_{k+1}) - L_{HGPO}(\theta_k) \leq -\frac{\alpha}{2}\|\nabla_\theta L_{HGPO}(\theta_k)\|^2 \tag{46}$$

This implies that whenever $\nabla_\theta L_{HGPO}(\theta_k) \neq 0$, the objective strictly decreases. If the gradient is zero, the objective no longer changes, indicating that the algorithm has reached a stationary point.

**$L$-Smoothness Discussion:** The components of $L_{HGPO}(\theta)$ are:

- $-L^{\text{POLICY}}(\theta)$: $L^{\text{POLICY}}(\theta)$ involves $\min$ and clipping operations, making it non-smooth at certain points. However, it is piecewise smooth in most regions, and gradient-based optimization remains effective in practice despite these non-smooth points.
- $c_1 S[\pi_\theta]$: The entropy term is smooth with respect to the policy parameters $\theta$.
- $L^{\text{HARM}}(\theta)$: Since $\bar{R}_g$ is the expectation of $r_t$, and $r_t$ depends on $\theta$ (through state/action selection and the temperature coefficient $\tau_u^{(t)}$), the variance $\text{Var}[\bar{R}_g]$ is a smooth quadratic function of $\bar{R}_g$. Therefore, $L^{\text{HARM}}(\theta)$ is also smooth.

### C.3.4 CONVERGENCE TO A STATIONARY POINT

We have already shown that $L_{HGPO}(\theta)$ is lower-bounded by $L_{HGPO,\text{min}}$, and that if the learning rate $\alpha_k$ is chosen suitably (e.g., $\alpha_k = \alpha < 1/L$), then:

$$L_{HGPO}(\theta_{k+1}) \leq L_{HGPO}(\theta_k) - \frac{\alpha}{2}\|\nabla_\theta L_{HGPO}(\theta_k)\|^2 \quad \text{(Eq. F)} \tag{47}$$

This implies that the sequence $\{L_{HGPO}(\theta_k)\}_{k \geq 0}$ is non-increasing.

Summing Eq. F from $k = 0$ to $N - 1$:

$$\sum_{k=0}^{N-1} (L_{HGPO}(\theta_{k+1}) - L_{HGPO}(\theta_k)) \leq -\frac{\alpha}{2} \sum_{k=0}^{N-1} \|\nabla_\theta L_{HGPO}(\theta_k)\|^2 \tag{48}$$

This gives:

$$L_{HGPO}(\theta_N) - L_{HGPO}(\theta_0) \leq -\frac{\alpha}{2} \sum_{k=0}^{N-1} \|\nabla_\theta L_{HGPO}(\theta_k)\|^2 \tag{49}$$

Rearranging:

$$\frac{\alpha}{2} \sum_{k=0}^{N-1} \|\nabla_\theta L_{HGPO}(\theta_k)\|^2 \leq L_{HGPO}(\theta_0) - L_{HGPO}(\theta_N) \tag{50}$$

Since $L_{HGPO}(\theta_N) \geq L_{HGPO,\text{min}}$, it follows that:

$$\frac{\alpha}{2} \sum_{k=0}^{N-1} \|\nabla_\theta L_{HGPO}(\theta_k)\|^2 \leq L_{HGPO}(\theta_0) - L_{HGPO,\text{min}} \tag{51}$$

The right-hand side is a finite constant. As $N \to \infty$, for the series $\sum_{k=0}^{\infty} \|\nabla_\theta L_{HGPO}(\theta_k)\|^2$ to converge, it must be that:

$$\lim_{k \to \infty} \|\nabla_\theta L_{HGPO}(\theta_k)\|^2 = 0 \tag{52}$$

and hence:

$$\lim_{k \to \infty} \|\nabla_\theta L_{HGPO}(\theta_k)\| = 0 \tag{53}$$

This proves that the gradient norm converges to zero, i.e., the algorithm converges to a stationary point $\theta^*$ where $\nabla_\theta L_{HGPO}(\theta^*) = 0$.

# D MULTI-PROTOTYPE ENHANCED GRAPH LEARNER

Given the user set $U$, the item set $I$, and the user-item interaction matrix $R \in \{0,1\}^{|U| \times |I|}$, a user-item bipartite graph $G = (V, E)$ can be constructed, where $V = U \cup I$ represents the nodes and $E = \{(u, i) \mid R_{ui} = 1\}$ represents the edges.

## D.1 GRAPH CONVOLUTIONAL LAYER

The graph convolution method of LightGCN He et al. (2020) is used for information propagation. The embedding calculation for the $l$-th layer of users $u$ and items $i$ is as follows:

$$z_u^{(l+1)} = \sum_{i \in N(u)} \frac{1}{\sqrt{|N(u)||N(i)|}} z_i^{(l)}, \quad z_i^{(l+1)} = \sum_{u \in N(i)} \frac{1}{\sqrt{|N(i)||N(u)|}} z_u^{(l)} \tag{54}$$

where $z_i^{(0)} = e_i$ represents the initial embedding after the previous method fuses the CoT feature, and $z_u^{(0)} = e_u$. $N(u)$ and $N(i)$ denote the neighbor sets of user $u$ and item $i$, respectively.

## D.2 FINAL EMBEDDING REPRESENTATION

After passing through $L$ layers of graph convolution, the final user and item embeddings are obtained by averaging the weighted embeddings from all layers:

$$z_u = \sum_{l=0}^{L} \alpha_l z_u^{(l)}, \quad z_i = \sum_{l=0}^{L} \alpha_l z_i^{(l)} \tag{55}$$

where $\alpha_l$ is the layer weight, typically set to $\frac{1}{L+1}$, and $z_u, z_i \in \mathbb{R}^{d'}$ are the final representations used for downstream tasks.

## D.3 INFONCE LOSS

The enhanced representation is obtained by aggregating the directly connected nodes. The contrastive loss for user $u$ and item $i$ is as follows:

$$L_{\text{st-user}} = -\sum_{u \in U} \ln \frac{\exp\left(\text{sim}(z_u^{(0)}, z_u^{(k)})/\tau\right)}{\exp\left(\text{sim}(z_u^{(0)}, z_u^{(k)})/\tau\right) + \sum_{v \in U, v \neq u} \exp\left(\text{sim}(z_u^{(0)}, z_v^{(k)})/\tau\right)} \tag{56}$$

$$L_{\text{st-item}} = -\sum_{i \in I} \ln \frac{\exp\left(\text{sim}(z_i^{(0)}, z_i^{(k)})/\tau\right)}{\exp\left(\text{sim}(z_i^{(0)}, z_i^{(k)})/\tau\right) + \sum_{j \in I, j \neq i} \exp\left(\text{sim}(z_i^{(0)}, z_j^{(k)})/\tau\right)} \tag{57}$$

Where $\text{sim}(z_u^{(0)}, z_u^{(k)}) = \frac{z_u^{(0)\top} z_u^{(k)}}{\|z_u^{(0)}\|\|z_u^{(k)}\|}$ is the cosine similarity, and $\tau$ is the temperature coefficient. $z_v^{(k)}$ is the embedding from other users (negative samples). The total contrastive loss is as follows:

$$L_{\text{struct}} = L_{\text{st-user}} + L_{\text{st-item}} \tag{58}$$

