# OpenReview forum: "LLM-CoT Enhanced Graph Neural Recommendation with Harmonized Group Policy Optimization"
_ICLR.cc/2026/Conference — ICLR 2026 Conference Withdrawn Submission_

### Official Review · Reviewer_TkSg · 2025-10-27

**Soundness:** 2
**Presentation:** 2
**Contribution:** 2
**Rating:** 2
**Confidence:** 5

**Summary:**

This paper proposes LGHRec, a framework that enhances graph neural network-based recommender systems through two main components: (1) Deep Semantic Embedding Generator (DSEG), which uses LLM Chain-of-Thought reasoning to generate semantic IDs for items offline, and (2) Harmonized Group Policy Optimization (HGPO), a reinforcement learning algorithm that optimizes graph contrastive learning via adaptive negative sampling and temperature coefficient adjustment. The method aims to address low information density in ID-based features, false negative sampling issues, and performance imbalances in long-tail recommendations. Experiments on three datasets show 3-7% improvements over baseline methods.

**Strengths:**

**Originality:**
- Combining LLM CoT reasoning with GNN-based recommendations is a reasonable idea
- The cross-group coordination mechanism in HGPO addresses a gap in GRPO
- Offline preprocessing strategy avoids online LLM inference latency

**Quality:**
- Extensive experiments across 9 baseline models and 3 datasets
- Detailed ablation studies examining different components
- Analysis of performance across different user/item activity levels (Figure 5)
- Convergence proof provided in appendix (though standard)

**Clarity:**
- Clear motivation illustrated in Figure 1
- Architecture diagram (Figure 2) shows overall framework
- Comprehensive appendix with implementation details

**Significance:**
- Addresses practical problems in recommender systems (long-tail items, contrastive learning optimization)
- Shows consistent but modest improvements (3-7%) across datasets
- Offline approach may be deployable in industrial settings

**Weaknesses:**

### 1. **Limited Technical Novelty **

The paper combines existing techniques without significant innovation:

**DSEG Component:**
- Using LLMs to generate text descriptions is standard practice
- CoT prompting (Figure 4) is straightforward application of existing techniques (Wei et al., 2022)
- Encoding with BERT is standard - **why not use the LLM's own embeddings?** This adds complexity
- Mixed fine-tuning to prevent catastrophic forgetting is well-known
- Simply concatenating semantic IDs with ID embeddings is the most basic fusion method

**HGPO Component:**
- Essentially GRPO + variance minimization term (Eq. 19)
- The reward function (Eqs. 1-6) is entirely hand-crafted with **9 hyperparameters** (θFN, θeasy, θFP, θeasy_low, w1-w5)
- No learning of reward structure - just manual threshold tuning
- Adaptive temperature via reinforcement learning has been explored in other contrastive learning contexts

### 2. **Weak Theoretical Contributions (Major)**

**Convergence Proof (Appendix C):**
- The proof is standard gradient descent convergence for smooth functions
- **Critical admission** (line 1010-1011): Authors acknowledge L-smoothness doesn't actually hold due to min/clipping operations in the loss
- Provides no novel theoretical insights specific to the problem
- Convergence to local minimum is expected and not a contribution

**Missing Theory:**
- Why does CoT reasoning improve recommendations? No formal analysis
- What properties of the reward structure guarantee good negative sample selection?
- Under what conditions does HGPO outperform GRPO?
- No generalization bounds or sample complexity analysis

### 3. **Experimental Weaknesses **

**Modest Improvements:**
- 3-7% improvements are marginal given the added complexity
- On MIND dataset (densest), improvements are smallest (2-7.49%), suggesting the method works best where it's needed least
- Standard deviations not reported for many results in Table 1 - are improvements statistically significant?

**Unfair Baseline Comparisons:**
- TALLRec and SPRec are "LLM-as-RS" methods solving a different problem (end-to-end generation)
- **Missing critical baselines**: Other LLM-enhanced methods that use semantic features (e.g., recent works using LLM embeddings for cold-start, knowledge-enhanced recommendations)
- No comparison with simpler alternatives: What if you just use pre-trained LLM embeddings without CoT? What about using GPT-style embeddings instead of BERT?

**Hyperparameter Sensitivity:**
- The reward function has 9 hyperparameters that require careful tuning (Table 6)
- Figure 8 shows performance is sensitive to λharm, c1, w5
- How do you set these in practice? Cross-validation on every new dataset?
- This makes the method impractical for real deployment

**Computational Cost:**
- Table 4 shows **15.2 hours** for LLM inference on Amazon-Book (903.7 min)
- Training time increases significantly (11.8s → 16.1s per epoch on Yelp)
- How often must items be re-processed? Items in recommender systems change frequently
- For industrial systems with millions of items, this is prohibitive

### 4. **Questionable Design Choices**

**Why BERT encoding?**
- The LLM (Qwen-2.5-32B) already produces embeddings
- Adding BERT encoding adds complexity and another model to maintain
- No justification for this design choice

**Concatenation fusion:**
- Simple concatenation + linear layer is the most basic fusion method
- Have you tried: attention-based fusion, gating mechanisms, learned weighted combination?
- Ablation (Table 2) shows weighted summation is worse, but many other options exist

**Mixed fine-tuning trade-off:**
- Figure 3 shows mixed fine-tuning prevents catastrophic forgetting
- But this means the LLM isn't really learning recommendation-specific CoT reasoning
- It's just maintaining general capabilities while adding domain examples
- Does this contradict the claim about leveraging deep CoT reasoning?

**CoT necessity:**
- The prompt (Figure 4) asks for 5 characteristics and sources
- Is this actually "reasoning" or just structured extraction?
- Ablation shows improvement over raw text (Table 2), but the gain is modest (2-4%)

### 5. **Long-tail Claims Not Well Supported **

The paper claims to improve long-tail recommendations:
- Figure 5 shows improvements across activity levels, but differences between LGHRec and baselines are often within error bars
- On MIND (right panel), improvements for low-activity users [5,10) appear minimal
- **No specific long-tail metrics**: Precision/Recall for cold-start items, coverage metrics, Gini coefficient
- Figure 7(c) shows variance reduction, but absolute performance for long-tail items is not directly evaluated

### 6. **Scalability and Practical Concerns **

**Item cold-start:**
- New items require LLM processing (15+ hours for 700K items)
- How do you handle new items in real-time?
- Periodic batch updates introduce staleness

**Hyperparameter tuning cost:**
- With 9 reward hyperparameters + 3 HGPO coefficients, tuning is expensive
- Table 6 shows sensitivity - requires careful tuning per dataset
- Industrial systems need methods that work out-of-the-box

**Memory requirements:**
- Table 4 shows modest memory increases
- But storing semantic IDs for millions of items adds overhead

**Questions:**

1. **Why BERT encoding instead of LLM embeddings?** The LLM already produces embeddings. Adding BERT seems to add complexity without clear justification. Can you show that BERT encoding outperforms using the LLM's own embeddings?

2. **Is CoT actually reasoning or structured extraction?** Your prompt (Figure 4) asks for 5 characteristics. This seems more like structured information extraction than deep reasoning. Can you demonstrate that the LLM is actually performing logical inference rather than just reformatting the input?

3. **How do you set 9+ hyperparameters in practice?** The reward function has θFN, θeasy, θFP, θeasy_low, w1-w5, plus λharm, c1, w5 for HGPO. How should practitioners set these? Grid search would be prohibitively expensive.

4. **Statistical significance?** Many improvements in Table 1 are 3-5% with no error bars. Are these statistically significant? Please provide significance tests.

5. **Comparison with simpler LLM-enhanced methods?** Why not compare with: (a) using pre-trained LLM embeddings without CoT, (b) using simpler text descriptions, (c) other recent LLM-enhanced recommendation works?

6. **Mixed fine-tuning contradiction?** Figure 3 shows mixed fine-tuning is best, but this prevents the LLM from specializing in recommendation CoT. Doesn't this undermine the claim about leveraging deep reasoning capabilities?

7. **Computational cost trade-offs?** 15 hours for preprocessing Amazon-Book is substantial. Can you provide cost-benefit analysis? At what scale does this become impractical?

8. **Long-tail specific evaluation?** Figure 5 shows activity-level breakdown, but can you provide dedicated cold-start metrics? Coverage? Gini coefficient? Evaluation specifically on tail items?

9. **Why does HGPO outperform GRPO?** The only difference is the variance minimization term (Eq. 19). Can you provide deeper analysis of why this helps? Is it just preventing over-optimization of popular items?

10. **Generalization across domains?** All three datasets are user-item interactions. Does this work for other types of recommendations (e.g., session-based, sequential with short sessions)?

---

### Official Review · Reviewer_4s6f · 2025-10-31

**Soundness:** 2
**Presentation:** 3
**Contribution:** 2
**Rating:** 2
**Confidence:** 3

**Summary:**

The paper proposes a plug-and-play module for GNN-based recommender systems that enables the incorporation of CoT reasoning from LLMs and also applies HGPO, Harmonized Group Policy Optimization, to optimize negative sampling in recommender systems. Outperformed experimental results on three widely used datasets validates the efficacy of the proposed method.

**Strengths:**

- The paper effectively leverages LLM-based high-quality representations through the generation of semantic IDs, which contributes to the observed performance gains.
- The experiment that groups users by interaction count and demonstrates robustness under long-tailed user distributions aligns well with the claimed contributions in the introduction, reinforcing the practical relevance of the proposed approach.
- The paper presents an effective ablation study, which thoroughly evaluates the contribution of each component in the proposed framework.

**Weaknesses:**

- While the authors conduct a sensitivity analysis, the method still involves a large number of hyperparameters that require careful tuning, which may limit the practical applicability and ease of deployment of the approach.
- Since Entropy Regularization loss is related to long-tail items, showing an ablation study on long-tailed items with this loss may enhance the novelty of the proposed method.
- Although random negative sampling is acknowledged as a limitation, many graph-based contrastive learning methods also face similar challenges. Demonstrating that even the latest contrastive learning approaches struggle with this issue could further highlight the significance and contribution of incorporating LLM-based representations in the proposed method.
- The visualization in Figure 6 is difficult to differentiate among the three methods.

**Questions:**

1. Could the authors justify more why the NDCG increases when applying mixed data compared to using only domain data?
2. During optimization, could the authors provide the reward trend to demonstrate that the reward consistently increases as intended by the model?
3. How are the positive sample embedding and the candidate negative sample pool defined?
4. Are the user and item embedding dimensions of the base model the same when incorporating LGHRec? Setting them to identical values would provide a fairer comparison, as some methods’ performance may depend on the embedding dimensionality.

---

### Official Review · Reviewer_mkMn · 2025-11-04

**Soundness:** 2
**Presentation:** 2
**Contribution:** 2
**Rating:** 4
**Confidence:** 4

**Summary:**

This paper proposes LGHRec, which fuses LLM step-by-step reasoning with reinforcement-learning-driven contrastive calibration to jointly upgrade semantic richness and training fairness for all user & item groups.

**Strengths:**

1. This work integrates LLMs and graph learning for recommendation, which is a novel topic that captures current trends and interests in the field.

2. The paper is well-written and easy to follow, with a clear articulation of the motivation behind the research.

3. The conducted experiments demonstrate the strengths of the proposed method, providing solid evidence of its effectiveness.

**Weaknesses:**

1. In terms of overall comparison, it appears that the authors only compare their proposed method against the base model. To further validate the effectiveness of their approach, they should consider including comparisons with other LLM-enhanced recommendation systems.

2. The authors should incorporate both significance analysis and case studies to provide a more comprehensive demonstration of the effectiveness of their work.

3. What is the efficiency of the proposed method? The authors should consider adding efficiency experiments in the draft.

**Questions:**

See Weakness.

---

### Official Review · Reviewer_UuYJ · 2025-11-04

**Soundness:** 2
**Presentation:** 1
**Contribution:** 2
**Rating:** 2
**Confidence:** 4

**Summary:**

The paper presents LGHRec, a framework that integrates the Chain-of-Thought (CoT) reasoning capabilities of Large Language Models (LLMs) with Graph Neural Network (GNN)-based recommender systems to enhance both semantic representation and contrastive learning. Specifically, the authors employ LLM-generated CoT reasoning to create high-density semantic ID embedding for items, which are encoded and fused with traditional ID embeddings, thereby enriching the representation space without incurring additional online costs. To address limitations in existing graph contrastive learning—particularly issues around fixed temperature coefficients and random negative sampling—the paper proposes Harmonized Group Policy Optimization (HGPO), a reinforcement learning strategy that adaptively selects negative samples and adjusts temperature values based on node characteristics, while ensuring cross-group consistency to mitigate performance disparities across long-tail and high-activity nodes. Extensive experiments across multiple datasets and models demonstrate consistent improvements, with NDCG@20 gains between 3% and 7%, especially in sparse data settings.

**Strengths:**

1. The paper proposes a new RecSys training paradigm that aims to solve the disadvantages of previous efforts.

2. Paper proposes deep semantic embedding generator that generate much richer information through fine-tuned LLM.

3. The paper designs a new reinforcement training algorithm based on grouped user/items.

**Weaknesses:**

See Questions.

**Questions:**

1. The paper is poorly written. For example, (1) authors spend too much words on describing the weakness of current methods in Abstract and Introduction. It is unnecessary and will shift away reader's attention to your main contribution. (2) Figure 1 is totally unnecessary for your paper. Authors do not need to explain the basic concept of LLM enhanced RecSys. (3) Most of figures in your paper are to vague to see, like Figure 3, 6, 7, 8.  The fonts on the figures are too small to observe.

2. Authors discussed all the disadvantages of previous methods from graph contrastive learning, LLM-based RecSys and negative sampling. Can your method solve all the mentioned disadvantages at the same time? If those are not the problems authors try to solve, why bother mention them?

3. Prompt grammatical error in Figure 4. "Each keyword must be should focus on the item's attributes."

4. There are so many thresholds in your method. How did authors decide them?

5. The method design seems really combinational with several seeming unrelated modules. In my point of view, find the root cause and solution for each one of them is more worth to investigate than just combine them together.

---

### Note · Authors · 2025-11-12

I have read and agree with the venue's withdrawal policy on behalf of myself and my co-authors.